# Single-nucleus transcriptomics reveals functional compartmentalization in syncytial skeletal muscle cells

Minchul Kim[1,4], Vedran Franke [2,4], Bettina Brandt[1], Elijah D. Lowenstein [1], Verena Schöwel[3], Simone Spuler [3], Altuna Akalin [2✉] & Carmen Birchmeier [1✉]

Syncytial skeletal muscle cells contain hundreds of nuclei in a shared cytoplasm. We investigated nuclear heterogeneity and transcriptional dynamics in the uninjured and regenerating muscle using single-nucleus RNA-sequencing (snRNAseq) of isolated nuclei from muscle fibers. This revealed distinct nuclear subtypes unrelated to fiber type diversity, previously unknown subtypes as well as the expected ones at the neuromuscular and myotendinous junctions. In fibers of the *Mdx* dystrophy mouse model, distinct subtypes emerged, among them nuclei expressing a repair signature that were also abundant in the muscle of dystrophy patients, and a nuclear population associated with necrotic fibers. Finally, modifications of our approach revealed the compartmentalization in the rare and specialized muscle spindle. Our data identifies nuclear compartments of the myofiber and defines a molecular roadmap for their functional analyses; the data can be freely explored on the MyoExplorer server (https://shiny.mdc-berlin.de/MyoExplorer/).

[1] Developmental Biology/Signal Transduction, Max Delbrueck Center for Molecular Medicine, Berlin, Germany. [2] Berlin Institute for Medical Systems Biology, Max Delbrueck Center for Molecular Medicine, Berlin, Germany. [3] Muscle Research Unit, Experimental and Clinical Research Center, Charité Universitätsmedizin Berlin and Max Delbrueck Center for Molecular Medicine, Berlin, Germany. [4] These authors contributed equally: Minchul Kim, Vedran Franke. ✉email: altuna.akalin@mdc-berlin.de; cbirch@mdc-berlin.de

All cells need to organize their intracellular space to properly function. In doing so, cells employ various strategies like phase separation, polarized trafficking, and compartmentalization of metabolites[1–4]. Syncytial cells face an additional challenge to this fundamental problem because individual nuclei in the syncytium can potentially have distinct functions and express different sets of genes. One interesting example is the skeletal muscle fiber, a syncytium containing hundreds of nuclei in a very large cytoplasm that possesses functionally distinct compartments. The best-documented compartment is located below the neuromuscular junction (NMJ), the synapse formed between motor neurons and muscle fibers. NMJ form in a narrow central region of the fiber, and are characterized by the enrichment of proteins that function in the transmission of the signal provided by motor neurons to elicit muscle contraction[5–9]. Motor neurons are known to instruct myonuclei at the synapse to express genes that function in synaptic transmission. Another specialized compartment is located at the end of the myofibers where they attach to the tendon, allowing force transmission. Many cell adhesion and cytoskeletal proteins are known to be enriched at the myotendinous junction (MTJ)[10,11]. However, little is known about the transcriptional characteristics of MTJ myonuclei and to date only a few genes like *Col22a1*, *Ankrd1,* and *LoxL3* were reported to be specifically expressed at the mammalian MTJ[12–14]. Previous studies have reported that the diffusion of transcripts and proteins in myofibers is limited, and indeed specific transcripts and proteins associated with the NMJ and MTJ appear to diffuse little inside the fiber[5,11,15,16]. Therefore, locally regulated transcription plays an important role in establishing functional compartments in the muscle. In addition, stochastic transcription of particular genes has been reported in myofibers, but it is unknown whether this reflects differences in myonuclear identities[17]. Since a systematic analysis is currently lacking, we neither know the extent of myonuclear heterogeneity nor can we assess whether additional myonuclear types exist beyond those at the NMJ and MTJ. Such knowledge may provide insight into how skeletal muscle cells orchestrate their many functions.

Previous studies on gene expression in the muscle relied on the analysis of selected candidates by in situ hybridization or on profiling the entire muscle tissue. The former is difficult to scale up, whereas the latter averages the transcriptomes of all nuclei. More recently, several studies have used single-cell approaches to reveal the cellular composition of the entire muscle tissue[18–20]. However, these approaches did not sample the syncytial myofibers. Single-nucleus RNA-Seq (snRNAseq) using cultured human myotubes failed to detect transcriptional heterogeneity among nuclei[21], underscoring the importance of studying the heterogeneity in an in vivo context where myofibers interact with surrounding cell types.

## Results

**Single-nucleus RNA-Seq analysis of uninjured and regenerating muscles.** We genetically labeled mouse myonuclei by crossing a myofiber-specific Cre driver (HSA-Cre) with a Cre-dependent H2B-GFP reporter. H2B-GFP is deposited at the chromatin, which allows us to isolate single myonuclei using flow cytometry. Nuclei of regenerating fibers were also efficiently labeled 7 days after cardiotoxin-induced injury (7 days post injury; 7 d.p.i.) (Supplementary Fig. 1a; note that nuclei in uninjured and regenerating fibers locate peripherally and centrally, respectively[22]). We confirmed the efficiency and specificity of the H2B-GFP labeling (Supplementary Fig. 1a–c). H2B-GFP was absent in endothelia (Cd31+), Schwann cells (Egr2+), tissue resident macrophages (F4/80+), and muscle stem cells (Pax7+)

(Supplementary Fig. 1c); nuclei of these diverse cell types lie outside the fiber and together make up around 50% of all nuclei in the tissue.

We next established a protocol for the rapid isolation of myonuclei. Conventional methods involve enzymatic dissociation of muscle fibers at 37 °C, which can cause secondary changes in gene expression[23–25]. We used a procedure that took 20 min from dissection to flow cytometry, employing fast mechanical disruption on ice. Indeed, our subsequent analysis indicated that this protocol avoided the expression of stress-induced genes (see Methods).

For snRNAseq profiling, we used the CEL-Seq2 technology[26], a low throughput plate-based method with high gene detection sensitivity[27]. Considering only exonic reads and genes detected in at least five nuclei, we detected 1000–2000 genes per nucleus (Supplementary Fig. 2a, b). Median mitochondrial read thresholds were 1.3% or less in all samples used in this study (Supplementary Fig. 2c). We analyzed nuclei from uninjured (1,591 nuclei) and regenerating *tibialis anterior* (TA) muscle (7 and 14 d.p.i., 946 and 1,661 nuclei, respectively). Uniform Manifold Approximation and Projection (UMAP) analysis of these datasets revealed heterogeneity among myonuclei (Fig. 1a). All nuclei expressed high levels of *Ttn*, a pan-muscle marker (Fig. 1b). The TA muscle contains three different fiber types (IIA—intermediate, IIB—very fast, and IIX—fast) that express distinct myosin genes. The largest cluster, bulk myonuclei, could be sub-divided into nuclei from distinct fiber types (Fig. 1b); *Myh1*- (IIX; lower left part of the cluster) or *Myh4* (IIB; upper right part of the cluster)-positive nuclei were most abundant and present roughly in a ratio of 1:1. *Myh2* (IIA)-expressing nuclei represented a minor population, consistent with the reported proportion of fiber types[28]. Notably, *Myh2* expressing nuclei mainly located to the *Myh1*-positive side in the UMAP plot, but not to the *Myh4*-positive side (Fig. 1b). By fluorescence in situ hybridization (FISH), we could readily observe Myh1/Myh2 co-expressing fibers, but not Myh2/Myh4 fibers (Supplementary Fig. 3).

Next, we defined genes that showed a dynamic expression profile during regeneration (Supplementary Fig. 4a and Supplementary Data 1). For instance, genes like *Arrdc2*, *Smox*, *Gpt2,* and *Pdk4* were strongly expressed in uninjured muscle but not during regeneration, whereas genes like *Mettl21c*, *Cish,* and *Slc26a2* were specifically expressed at 7 d.p.i. These results were validated by RT-qPCR using isolated GFP+ myonuclei and FISH (Supplementary Fig. 4b, c).

In addition to the bulk myonuclear population, we detected smaller populations with very distinct transcriptomes that expressed pan-muscle genes like *Ttn* (Fig. 1a, c). Like the bulk nuclei, distinct nuclei in these populations expressed different myosin genes, indicating that the heterogeneity is not driven by fiber type differences (Fig. 1b). We first searched for and identified a cluster specifically expressing known NMJ marker genes such as *Chrna1*, *Prkar1a, Ache,* and *Chrne* that was present in uninjured and regenerating muscle[5] (Fig. 1d and Supplementary Data 2). Our data identified many other genes not previously known to be specifically expressed at the NMJ such as *Vav3*, *Ablim2*, *Phldb2,* and *Ufsp1* (Fig. 1d). FISH of such markers and the known marker *Prkar1a* confirmed their specific expression at the NMJ (Fig. 1e). The full list of marker genes identified in this study is available in Supplementary Data 3.

**Two distinct nuclear populations at the myotendinous junction.** We found two clearly distinct nuclear populations that expressed MTJ-related genes in uninjured and regenerating muscle, and designated them MTJ-A and MTJ-B (Fig. 2a and

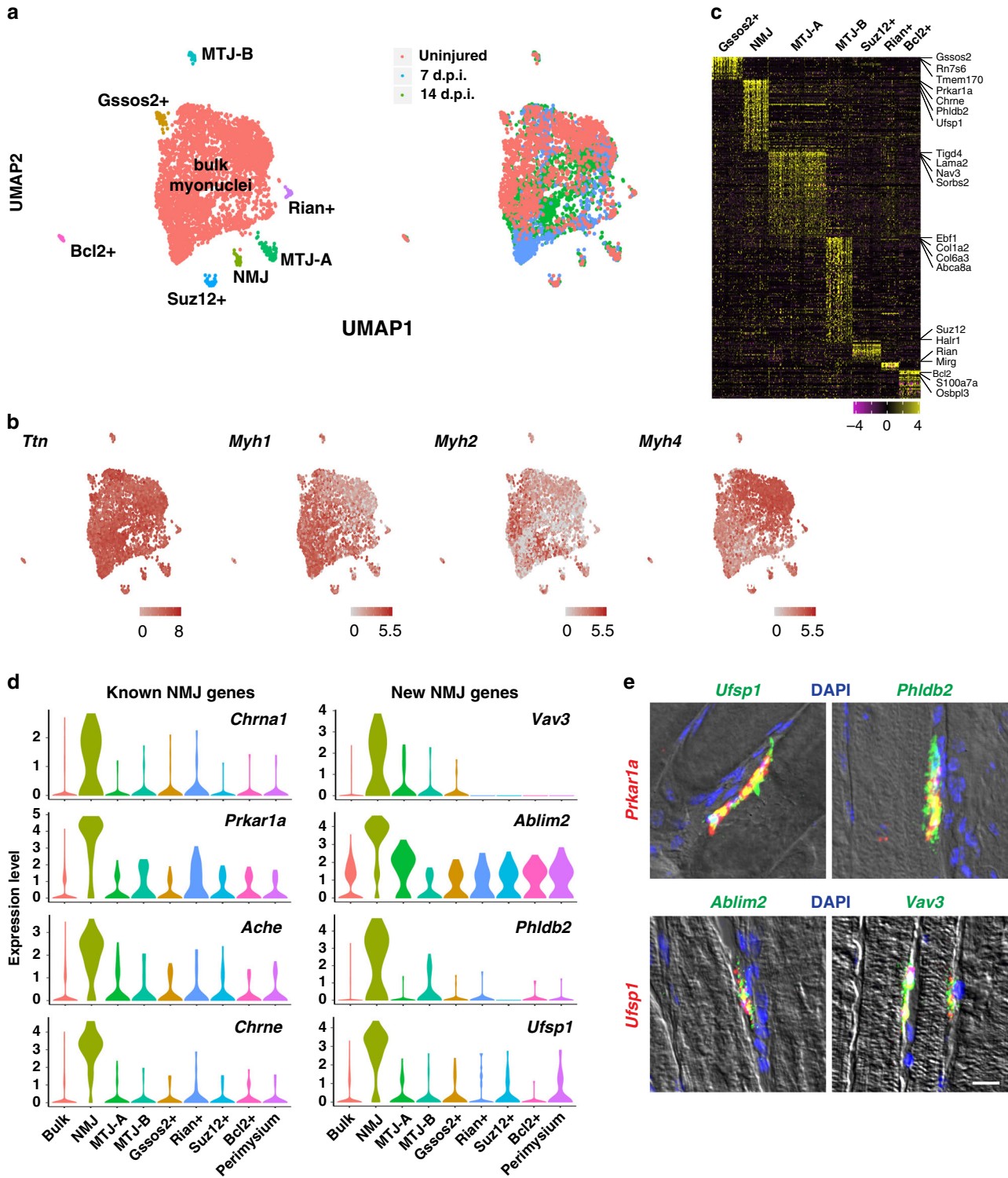

**Fig. 1 Nuclear heterogeneity in uninjured and regenerating muscles. a** UMAP plot of transcripts detected in nuclei of uninjured and regenerating muscles. The colors identify different nuclear populations (left) or nuclei in uninjured or regenerating muscle (right). The perimysium population is identified after re-clustering the bulk myonuclei and described further in Fig. 3. **b** Expression of *Ttn* and *Myosin* genes identifies myonuclei. **c** Heat-map of specific genes enriched in clusters other than the bulk myonuclei. Top representative genes are indicated on the side. **d** Violin plots of the previously known or NMJ marker genes identified here. **e** Upper row—conventional FISH against a known NMJ marker (*Prkar1a*) and NMJ genes identified here (green). Bottom row —single-molecule FISH against *Ufsp1* (red) and other newly identified NMJ genes (green). Expression patterns were validated in two or more individuals. Scale bar, 10 μm.

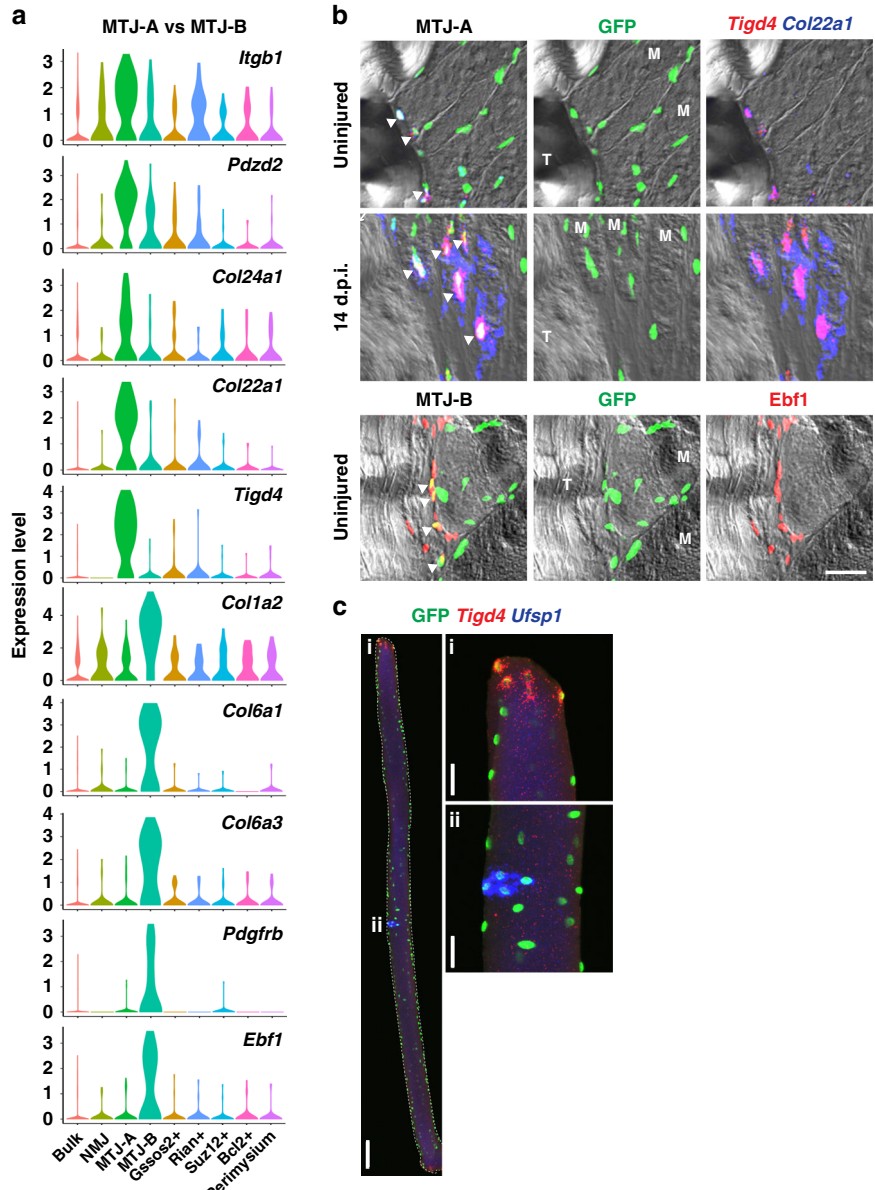

**Fig. 2 Two distinct nuclear populations at the myotendinous junction. a** Marker genes enriched in MTJ-A or MTJ-B nuclei are presented by violin plots. **b** Upper two rows—single-molecule FISH of two MTJ-A markers (*Tigd4* and *Col22a1*) in uninjured or 14 d.p.i TA muscle expressing H2B-GFP in myonuclei. Bottom row—Ebf1 (MTJ-B marker) immunofluorescence in uninjured TA muscle expressing H2B-GFP in myonuclei. Shown are MTJ regions; T, tendon and M, myofiber. Arrowheads indicate co-localization of MTJ marker genes and GFP. Scale bar, 30 μm. **c** Conventional FISH experiment in an isolated single EDL fiber. Insets show magnification of MTJ (i) and NMJ (ii) regions. Scale bars, 100 μm (for the entire fiber) and 30 μm (for the insets). Expression patterns were validated in two or more individuals.

Supplementary Data 2). MTJ-A nuclei expressed genes whose protein products are known to be enriched at the MTJ (e.g. *Itgb1*)[29] as well as specific collagens (e.g. *Col24a1* and *Col22a1*). *Col22a1* has been functionally characterized in zebrafish using morpholino knockdowns that disrupt MTJ formation[30]. MTJ-B nuclei expressed an alternative set of collagens that are known to be deposited at the MTJ such as *Col1a2*, *Col6a1*, and *Col6a3*[31]. *Col6a1* expression was particularly notable because its mutation causes Bethlem myopathy, which is characterized by deficits at the MTJ[32].

We validated the two top marker genes of MTJ-A (*Tigd4* and *Col22a1*) using FISH and observed their expression in nuclei at fiber endings (Fig. 2b and Supplementary Fig. 5). These transcripts were exclusively expressed from H2B-GFP positive myonuclei and present only at the MTJ. Their expression became

much more pronounced at 14 d.p.i. compared to uninjured muscle (Fig. 2b). To visualize heterogeneity within the syncytium, we isolated single fibers and performed double FISH (Fig. 2c). *Tigd4* FISH signals were detected at fiber ends where the MTJ is located, whereas *Ufsp1* transcripts appeared at the middle of the fiber where the NMJ is located.

We detected transcripts of MTJ-B genes (*Pdgfrb*, *Col6a3*) expressed from H2B-GFP nuclei at the MTJ in both uninjured muscle and at 14 d.p.i. (Supplementary Fig. 6). We also confirmed that Ebf1 protein is present in H2B-GFP positive myonuclei close to the MTJ (Fig. 2b). *Pdgfrb* and *Col6a3* are known to be expressed by the connective tissue, and indeed these transcripts were also detected in cells located distally to the MTJ and outside the fiber (Supplementary Fig. 6). However, such cells were neither marked by H2B-GFP nor by *Ttn*. Thus, MTJ-B nuclei co-

expressing muscle genes and *Pdgfrb*, *Col6a3,* or Ebf1 were exclusively located at end of the muscle fibers. Unlike MTJ-A nuclei, those expressing the MTJ-B signature were not found in every fiber. Because the MTJ-B signature includes both markers of muscle fibers and of connective tissue cells, we compared the gene signatures of MTJ-B nuclei to the ones of known cell types in the muscle tissue, specifically with the connective tissue cell types identified in a previous single-cell sequencing experiment that excluded syncytial myofibers[18]. None of these cell types expressed the MTJ-B signature. Thus, MTJ-B represents a nuclear population in the myofiber that co-expresses genes typical of the myofiber (e.g. *Ttn*) and of connective tissue (*Pdgfrb*, *Col6a3*, *Ebf1*).

**Identification of additional myonuclear populations**. Further previously unknown myonuclear compartments were identified by our systematic analysis, and we show exemplary genes preferentially expressed by each population in Fig. 3a. The first of these we named after the top marker, *Rian*, a maternally imprinted lncRNA. This cluster also expressed other lncRNAs that are all located at the same genomic locus (also known as *Dlk1-Dio3* locus) like *Mirg* and *Meg3* (Fig. 3a, b and Supplementary Data 3). The *Dlk1-Dio3* locus additionally encodes a large number of microRNAs expressed from the maternal allele, among them microRNAs known to target transcripts of mitochondrial proteins encoded by the nucleus[33]. FISH against *Rian* transcripts showed clear and strong expression in a subset of myonuclei (Fig. 3c). FISH on isolated fibers showed dispersed localization of *Rian* expressing nuclei without clear positional preference (Fig. 3d). A previous study reported that the *Dlk1-Dio3* locus becomes inactive during myogenic differentiation[34]. However, our results show that some myonuclei retain expression, which might be important for the metabolic shaping of the fiber.

The top marker of the second cluster was *Gssos2*, an antisense lncRNA, and these nuclei expressed many genes that function in endoplasmic reticulum (ER)-associated protein translation and trafficking (Fig. 3a, b). Among these were *Tmem170a* and *Rab40b*. Tmem170a induces formation of ER sheets, the site of active protein translation[35], and Rab40b is known to localize to the Golgi/endosome and regulates trafficking[36]. Furthermore, the srpRNA (signal recognition particle RNA) *Rn7s6*, an integral component of ER-bound ribosomes, was markedly enriched in this population (Fig. 3a, b). The enrichment of srpRNA was also observed when the expression of repeat elements was quantified (Supplementary Fig. 7). FISH showed that *Gssos2* displayed a heterogeneous and strong expression in a subset of myonuclei and that *Rian* and *Gssos2* were expressed in different nuclei; we examined more than 100 Rian+ or Gssos2+ nuclei from 2 individuals and did not observe co-expressing nuclei. (Fig. 3c; Supplementary Fig. 8a). *Rian* and *Gssos2* were located away from NMJ nuclei (Supplementary Fig. 8b). Therefore, Rian+ and Gssos2+ nuclei represent independent nuclear populations.

Two remaining populations (Suz12+ and Bcl2+ nuclei) need further characterization. The top two markers expressed by Suz12+ nuclei were *Suz12*, a core Polycomb complex component, and *Halr1*, a long non-coding RNA expressed from the *Hoxa* locus, suggesting that specific mechanisms of epigenomic regulation might be used in these nuclei. Bcl2+ nuclei strongly expressed genes involved in steroid signaling such as *Osbpl3* (oxysterol-binding protein) and *Nr2f1* (steroids-sensing nuclear receptor).

Re-clustering of the bulk myonuclei in Fig. 1a revealed an additional nuclear subpopulation (Fig. 3e and Supplementary Fig. 9). This subpopulation was characterized by the enrichment of marker genes such as *Muc13* and *Gucy2e* (Fig. 3a, e). FISH

showed that myonuclei expressing *Muc13* were always located at the very outer part of the muscle tissue near the perimysium (Fig. 3f). A previous ultrastructural study suggested that myofibers and perimysium establish specialized adhesion structures[37], and our data suggest that we have detected a myonuclear compartment participating in this process.

**snRNAseq of fibers in *Mdx* dystrophy model**. To begin to understand whether and how myonuclear heterogeneity is altered in muscle disease, we conducted snRNAseq on *Mdx* fibers (1939 nuclei), a mouse model of muscular dystrophy caused by mutation of the *Dystrophin* gene (Fig. 4a and Supplementary Fig. 10a). To examine how the transcriptome of *Mdx* myonuclei is related to those of uninjured and regenerating muscle, we calculated gene signature scores of each nucleus based on the top 25 genes that distinguish uninjured and regenerating fibers. This showed that nuclei of the *Mdx* muscle mostly resembled those from the uninjured and 14 d.p.i muscle, whereas the signature of 7 d.p.i myofibers was depleted (Fig. 4b). Cluster A displayed marker genes that were largely shared with those specific to uninjured fibers like *Arrdc2*, *Glul*, *Smox,* and *Gpt2* (Fig. 4c and Supplementary Fig. 4a) and might correspond to nuclei from fibers that are little damaged or undamaged.

We found nuclear populations in the *Mdx* dataset that were not identified in the uninjured/regenerating muscle. The first of these highly expressed various non-coding transcripts, and further experiments demonstrated that nuclei expressing these transcripts were located inside dying fibers (Fig. 4c and Supplementary Fig. 10b). In particular, staining with mouse IgG that identifies fibers with leaky membranes demonstrated these ncRNAs were expressed in IgG+ fibers (Fig. 4d). Further, fibers strongly expressing these ncRNAs were highly infiltrated with H2B-GFP negative cells likely corresponding to macrophages (Supplementary Fig. 10c, d). In line with the idea that these nuclei represent dying fibers, they had low UMI counts suggesting low transcriptional activity or high mRNA degradation (bottom histogram in Fig. 4c). Whether the ncRNAs are the consequence or active contributors to fiber death needs further investigation. In addition, three populations (B1-B3) located adjacent to each other in the UMAP map had low UMI counts, but did not display any clear marker genes. We speculate that these nuclei might also originate from damaged fibers.

Next, we searched for the clusters identified in the uninjured and regenerating muscle. In the UMAP of the *Mdx* dataset, clusters corresponding to NMJ, MTJ-A, MTJ-B, Rian+, Gssos2+, and Bcl2+ populations were not identifiable. However, we detected two subpopulations also present in uninjured/regenerating muscle, Suz12+ and perimysial nuclei, pointing to some degree of specificity (Fig. 4a). We therefore used gene signature scores to identify nuclei that display a correlative expression of signature genes. Such inspection showed that MTJ-A nuclei were present in the UMAP but did not cluster together (Supplementary Fig. 10e). Nevertheless, marker genes of MTJ-A (*Tigd4* and *Col22a1*) robustly labeled the MTJ of *Mdx* muscle (Supplementary Fig. 10f). We speculate that fiber-level heterogeneity (e.g. dying fibers, intact fibers, and regenerating fibers) drives the shape of the UMAP map in *Mdx*, which might interfere with the clustering of MTJ-A nuclei. Unlike MTJ-A, nuclei with high signature scores of NMJ, MTJ-B, Rian+, Gssos2+, and Bcl2+ nuclei were not detected. We investigated the expression of NMJ genes in further depth. This showed that the strict co-expression of two NMJ marker genes (*Ufsp1* and *Prkar1a*) typical for the control muscle was lost, and that these genes were instead expressed in a dispersed manner in the *Mdx* muscle (Supplementary Fig. 10f). Notably, the histological structure of the NMJ

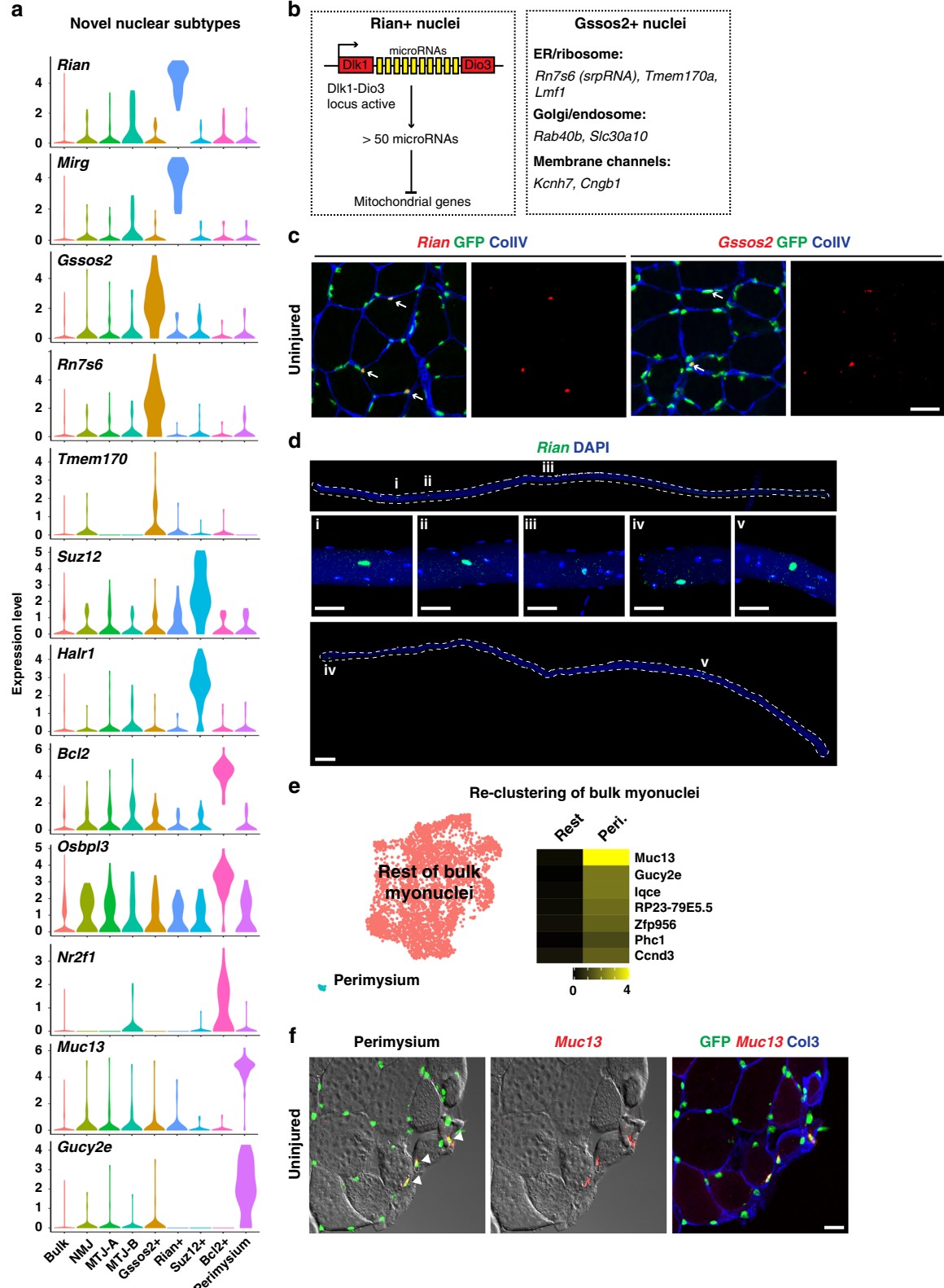

is known to be fragmented in *Mdx* mice[38,39], and our data suggest that also postsynaptic nuclei are incompletely specified.

### Emergence of a nuclear population implicated in fiber damage repair in *Mdx* model.

Marker genes of the cluster 'fiber repair' in Fig. 4a showed enrichment of ontology terms related to human

muscle disease (Fig. 4e). Indeed, many top marker genes were previously reported to be mutated in human myopathies (*Flnc, Klhl40,* and *Fhl1*)[40–42] or to directly interact with proteins whose mutation causes disease (Ahnak interacts with dysferlin; Hsp7b or Xirp1 interact with Flnc)[43–45]. Combinatorial FISH in tissue sections confirmed co-expression of such marker genes in a subset of nuclei of *Mdx* muscle, but such nuclei were not present

**Fig. 3 Identification of additional nuclear subtypes. a** Marker genes enriched in each of the nuclear populations are presented by violin plots. **b** Illustration of potential functions of Rian+ and Gssos2+ nuclei. Rian+ nuclei might regulate local mitochondrial metabolism through microRNAs embedded in *Dlk1-Dio3* locus, whereas Gssos2+ nuclei potentially regulate local protein synthesis and entry into the secretory pathway. **c** Validation of the top markers of Rian+ and Gssos2+ nuclei by single-molecule FISH in uninjured muscles. Note their strong expression in a subset of myonuclei (arrows). Scale bar, 30 μm. **d** Expression of *Rian* in isolated EDL fibers. Insets show magnifications of indicated regions. Scale bars, 100 μm (for entire fibers) and 30 μm (for insets). **e** (Left) UMAP plot of re-clustered bulk myonuclei identified in Fig. 1a. (Right) Heat-map showing differentially expressed genes in perimysium (peri.) nuclei versus rest of the bulk myonuclei. Averaged gene expression levels are shown for each gene. **f** Validation of *Muc13* expression in myonuclei adjacent to the perimysium by single-molecule FISH in uninjured muscle. Scale bar, 30 μm. Expression patterns were validated in two or more individuals.

in control muscle (Fig. 4f, Supplementary Fig. 11a, b). Further, nuclei expressing these genes were frequently closely spaced in fibers. We also verified that *FLNC* and *XIRP1* were co-expressed in nuclei from patient biopsies with confirmed *DYSTROPHIN* mutation, but not in healthy human muscle (Fig. 4g and Supplementary Fig. 11a).

Previous studies have established that Flnc and Xirp1 proteins localize to sites of myofibrillar damage to repair such insults[44,46,47], whereas Dysferlin, an interaction partner of Ahnak, functions during repair of muscle membrane damage[48]. Our analysis shows that these genes are transcriptionally co-regulated which might occur in response to micro-damage. To substantiate that this signature is not specific to muscular dystrophy caused by *Dystrophin* mutation, we investigated whether they can be identified in *Dysferlin* deficient muscle where the continuous micro-damage to the membrane is no longer efficiently repaired[48]. Again, we observed nuclei co-expressing *Flnc* and *Xirp1* in this mouse disease model and in biopsies from human patients with *DYSFERLIN* mutations (Supplementary Fig. 11c, d). We propose that the genes that mark this cluster represent a 'repair' signature. Notably, the accompanying paper[49] identified a similar population during late postnatal development and in the aging muscle, indicating that the 'repair' genes might also function during fiber remodeling.

Finally, we identified another previously unknown population in the *Mdx* muscle, cluster C, that expressed marker genes such as *Gpatch2*, *Emilin1,* and *Pde6a* not previously studied in a muscle context (Fig. 4c and Supplementary Data 3). The role of this population in muscle pathophysiology needs further characterization.

**Nuclear heterogeneity in muscle spindle fibers.** In principle, our approach can be used to explore nuclear heterogeneity in specific fiber types. We thus aimed to investigate heterogeneity in muscle spindles that detect muscle stretch and function in motor coordination[50]. Muscle spindles are very rare and ~10 spindles exist in a TA muscle of the mouse[51]. They contain bag and chain fibers, and their histology suggests further compartmentalization (Fig. 5a). *HSA-Cre* labels myonuclei of the spindle (Fig. 5b), but the overwhelming number of nuclei derive from extrafusal fibers. To overcome this, we used *Calb1-Cre* to specifically isolate spindle myofiber nuclei (Fig. 5b) and discovered different nuclear subtypes inside these specialized fibers (Fig. 5c).

Bag fibers are slow fibers and express *Myh7b*[52,53], whereas chain fibers are fast. A cluster expressing *Myh7b* and *Tnnt1*, a slow type troponin isoform, was assigned to mark Bag fibers. In contrast, two clusters expressed *Myh13*, a fast type Myosin, or *Tnnt3*, a fast type troponin, which we named Chain1 and Chain2. Strikingly, we identified a cluster that expressed a set of genes largely overlapping with those identified in NMJ nuclei of extrafusal fibers, e.g. *Chrne*, *Ufsp1,* and *Ache*, which we assign as the NMJ of the spindle (spdNMJ) (Fig. 5d, e). Furthermore, the spindle myotendinous nuclei (spdMTJ) expressed a significantly overlapping set of genes as those identified in MTJ-B nuclei of extrafusal fibers (Fig. 5d, e). MTJ-A markers were not detected.

Notably, the clusters Bag and spdNMJ expressed the mechanosensory channel *Piezo2*[54]. To verify the assignment and to define the identity of an additional large compartment (labeled as Sens), we validated the expression of different marker genes in H2B-GFP positive fibers of *Calb1-Cre* muscle in tissue sections (Supplementary Fig. 12b) and in fibers after manual isolation (Fig. 5f). FISH of *Calcrl*, a marker of the Sens cluster, showed specific localization to the central part of spindle fibers containing densely packed nuclei, the site where sensory neurons innervate (Fig. 5f). In the same fiber, transcripts of the spdNMJ marker gene *Ufsp1* located laterally as a distinct focus. In contrast, *Piezo2* was expressed throughout the lateral contractile part of the fiber, but was excluded from the central portion. Thus, the central non-contractile part of the muscle spindle that is contacted by sensory neurons represents a fiber compartment with specialized myonuclei clearly distinguishable from the spdNMJ.

**Profiling transcriptional regulators across distinct compartments.** To gain insights into the transcriptional control of the different nuclear compartments, we investigated the expression profile of transcription factors and epigenetic regulators (Fig. 6a and Supplementary Data 4). Notably, the transcript encoding Etv5 (also known as Erm), a transcription factor known to induce the NMJ transcriptome[55], and its functional homolog Etv4 were enriched in NMJ nuclei. Irf8 (3rd rank factor in NMJ) is also interesting as mutation of an Irf8 binding site in the *CHRNA* promoter causes *CHRNA* misexpression in the thymus and leads to the autoimmune disease myasthenia gravis[56], implicating Irf8 in the control of an NMJ gene in a tissue outside of the muscle. In MTJ-A nuclei, Smad3, the effector of TGF-β signaling, was found as the second rank factor. In addition, TGF-β receptors were also expressed by MTJ-A myonuclei. TGF-β is released by force from tenocytes and is required to maintain tenocytes[57], but our dataset suggests that MTJ myonuclei can also receive TGF-β signals.

To test whether our dataset can identify a functionally relevant factor, we chose to further study Ebf1, the most strongly enriched transcription activator in MTJ-B. ChIP-Seq data of Ebf1 (ENCODE project ENCSR000DZQ) showed that Ebf1 directly binds to ~70% of the genes we identified as MTJ-B markers (Fig. 6b). In contrast, Ebf1 binds less than 30% of NMJ or MTJ-A marker genes. We generated a C2C12 cell line in which Ebf1 expression was induced by doxycycline (Fig. 6c). RT-qPCR analysis of selected markers showed that many of them were induced in a dose-dependent manner upon Ebf1 over-expression (Fig. 6d), as was ColVI protein (Fig. 6c). To validate these findings in vivo, we analyzed *Ebf1* mutant muscle. Using FISH for *Ttn* to identify myonuclei, we observed a strong reduction of *Col6a3* and *Fst1l* transcripts in the *Ebf1* mutant compared to control MTJ myonuclei (Fig. 6e). Their expression from non-myonuclei that also express Ebf1 was also diminished. In contrast, the expression of NMJ, MTJ-A, and *Rian* markers was not affected in *Ebf1* mutants (Supplementary Fig. 13). Taken together, our dataset provides a template for the identification of regulatory factors that establish or maintain these compartments.

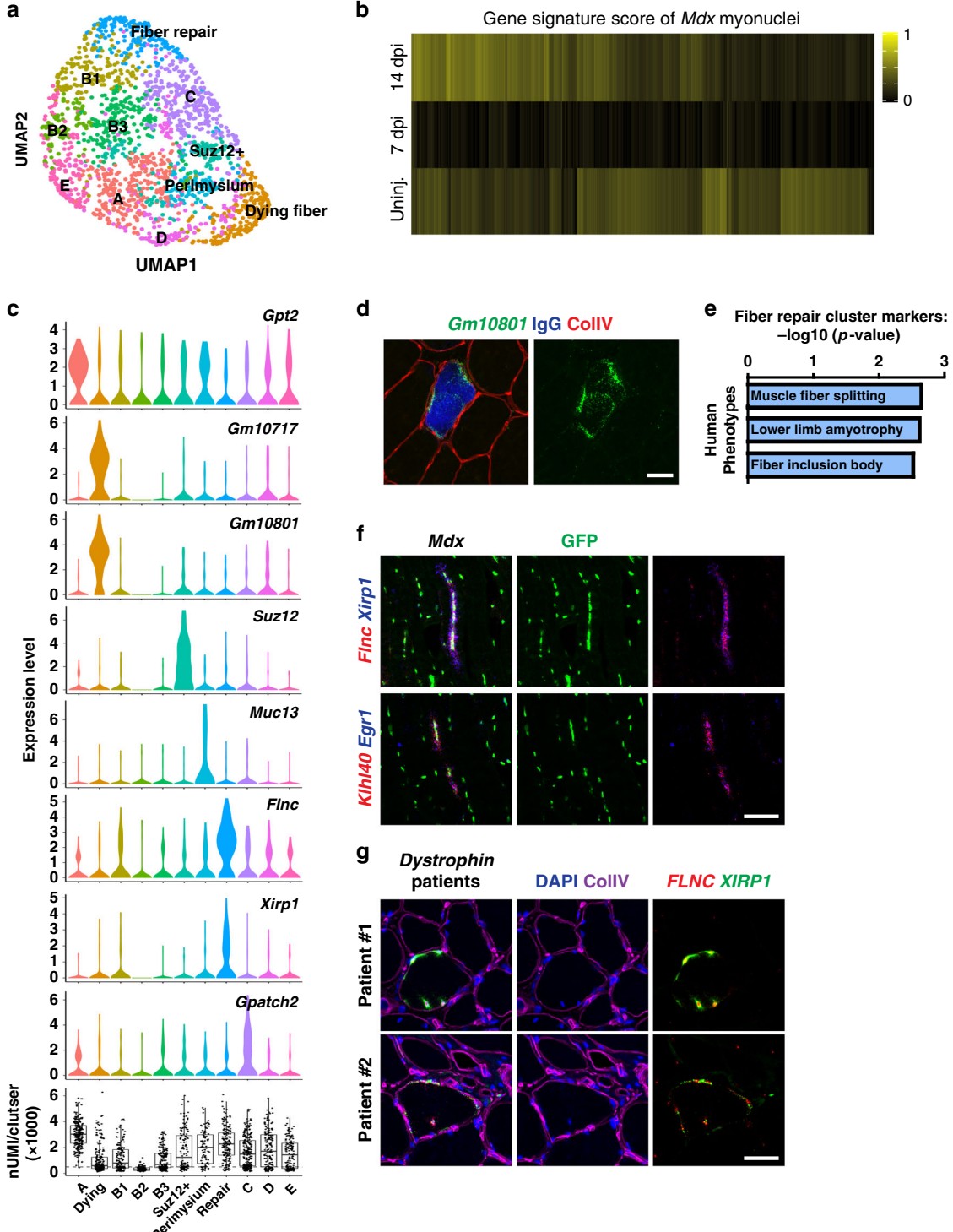

**Fig. 4 snRNAseq analysis of *Mdx* muscle. a** UMAP plot of *Mdx* myonuclei (1939 nuclei). **b** Each *Mdx* myonucleus was assigned a gene signature score, i.e. a gene expression score indicating similarity with uninjured (uninj.), 7 or 14 d.p.i. myonuclei. Each column represents an individual nucleus. **c** Marker genes enriched in each population are presented by violin plots. The bottommost histogram shows nUMI in each population. **d** The ncRNA *Gm10801* is expressed in IgG-positive fibers, indicating that it defines nuclei of necrotic fibers. **e** Fiber repair myonuclei express high levels of various genes implicated in fiber repair that are implicated in myopathies. The p-values were calculated by hypergeometric test and then were corrected for multiple comparisons using the Benjamini–Hochberg (BH) procedure. **f** Indicated marker genes of the cluster in (**e**) are co-expressed in longitudinal muscle sections of *Mdx* mice. **g** Co-expression of *FLNC* and *XIRP1* in the muscle from dystrophy patients carrying mutations in the *DYSTROPHIN* gene (*DYS* del exons15-18; *DYS* c.2323A>C). Control images for (**f**) and (**g**) are shown in Supplementary Fig. 11a. All scale bars, 50 μm. Expression patterns were validated in two or more individuals.

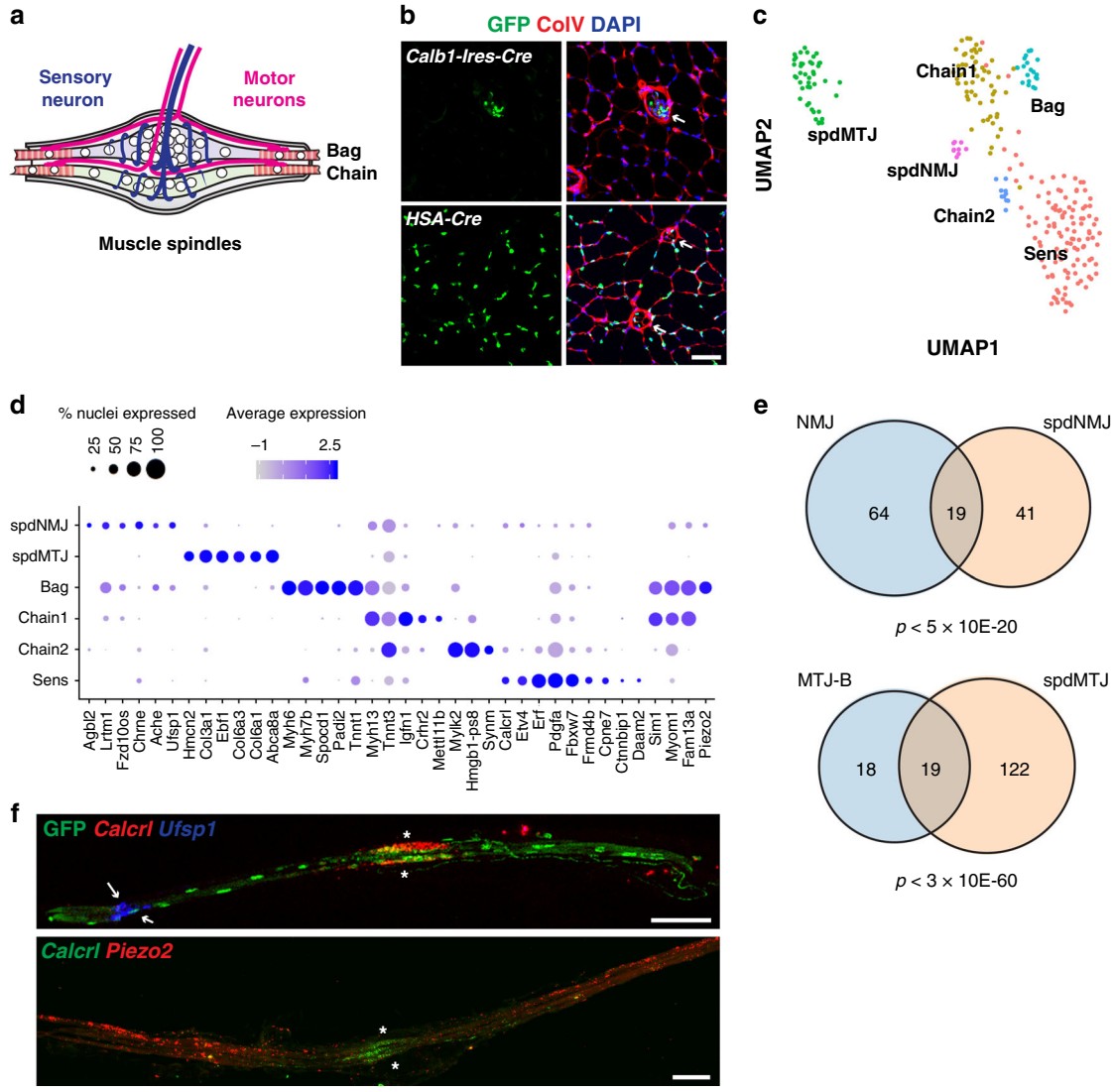

**Fig. 5 Functional compartments inside muscle spindle fibers. a** Schema showing the structure of the muscle spindle. **b** Specific labeling of spindle myonuclei using *Calb1-Ires-Cre*. Arrows indicate muscle spindles. **c** UMAP plot of muscle spindle myonuclei (260 nuclei). **d** Expression map of nuclear populations identified in (**c**). **e** Venn diagram comparing "spdNMJ vs extrafusal fiber NMJ" and "spdMTJ vs extrafusal fiber MTJ-B". Genes enriched in each population (average logFC > 0.7) were used to generate the diagrams. Statistical analysis was performed using hypergeometric test using all genes detected in uninjured/regenerating and spindle datasets as background. **f** Single-molecule FISH experiments of isolated muscle spindle fibers. Arrows indicate spdNMJ, and asterisks the central non-contractile parts of spindle fibers. All scale bars, 50 μm. Expression patterns were validated in two or more individuals.

## Discussion

Here, we used snRNAseq to systemically characterize the transcriptional heterogeneity of myofiber nuclei. Common to all was the expression of muscle-specific genes like *Ttn*, but small sub-populations were detected that expressed an additional layer of distinct and characteristic genes. Our analysis of the uninjured and regenerating muscle identified nuclear populations at anatomically distinct locations such as the NMJ and MTJ-A populations that were known to exist, as well as MTJ-B and perimysial nuclei, two populations that are first described here. Moreover, we found a number of populations that are scattered throughout the myofiber, among them the two distinct Rian+ and Gssos2+ nuclear subtypes. Thus, myonuclear populations are not always associated with distinctive anatomical features. How these nuclear subtypes emerge, i.e. whether they are associated with other cell types in the muscle tissue or arise stochastically needs further study. The functional role of the nuclear populations will need further characterization in the future. Collectively, our data

identified many genes that are specifically expressed in the various nuclear populations, providing a comprehensive resource for studying these compartments. We provide a webserver where users can freely explore the expression profile of their gene of interest in myonuclei (https://shiny.mdc-berlin.de/MyoExplorer/). Together, our results reveal the complexity of the regulation of gene expression in the syncytium and show how regional transcription shapes the architecture of multinucleated skeletal muscle cells.

Our analysis also shows that the transcriptional heterogeneity in myonuclei is dynamic. For instance, during regeneration the gene expression signatures of bulk nuclei differ from those of the uninjured muscle, and differences between early and later stages of regeneration can be detected. Further, the frequency of different myonuclear subtypes might suggest dynamic changes in nuclear compartments during regeneration (Supplementary Data 2). However, the proportion of these nuclear subtypes is low (0.5~3%) which precludes a definitive conclusion at this stage.

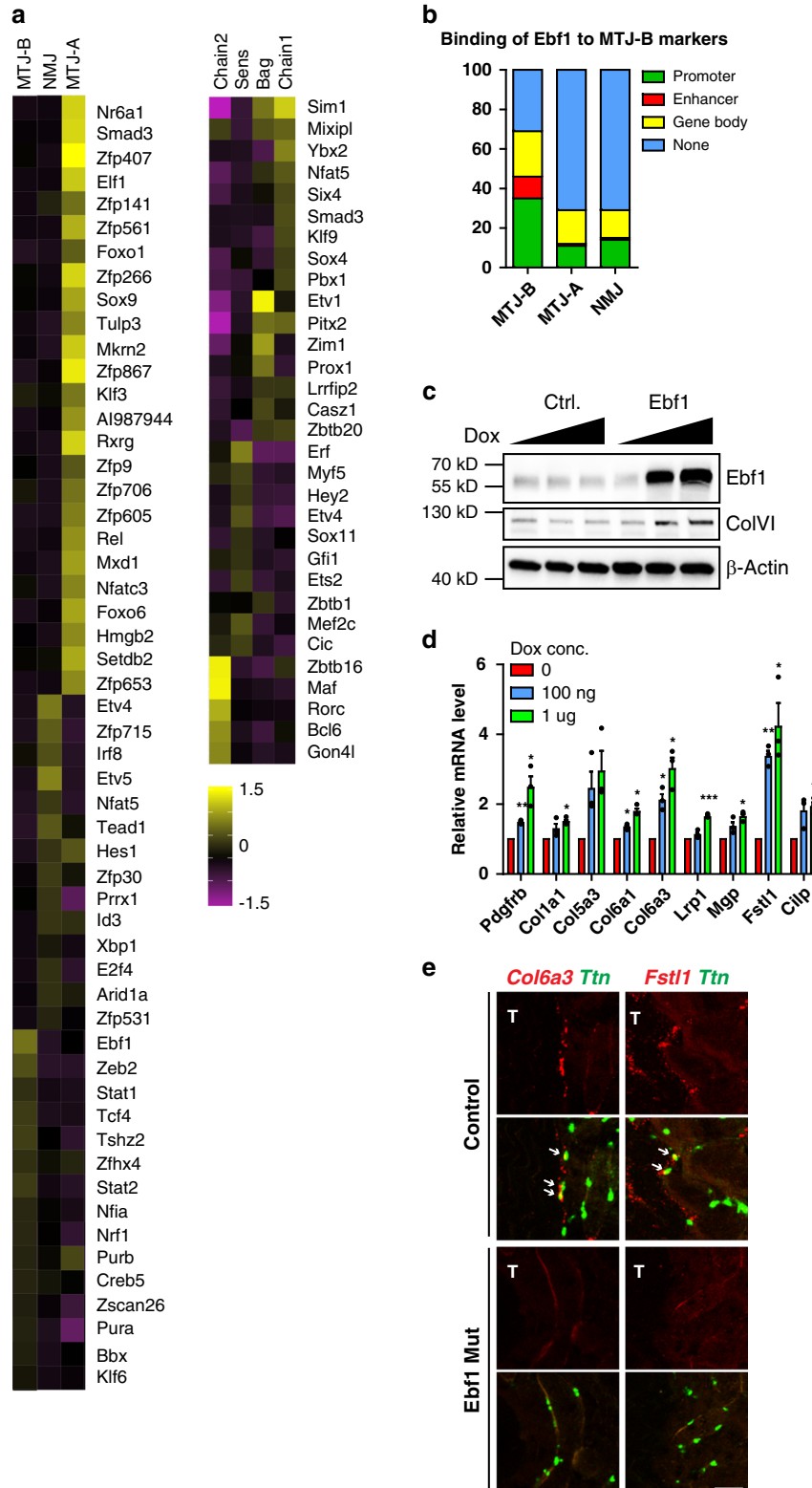

**Fig. 6 Expression profiles of transcription factors and epigenetic regulators across distinct nuclear subtypes. a** Heat-map showing the expression level of transcription factors and epigenetic regulators in indicated nuclear subtypes; see also Supplementary Data 4 for the full list including nuclear subtypes not shown in this figure. **b** Ebf1 directly binds to genomic regions of the top 100 genes identified to be specifically expressed in MTJ-B nuclei (ENCODE project ENCSR000DZQ), but much less to markers of MTJ-A or NMJ. Classification of the Ebf1 binding sites in these 100 genes. **c** Western blot analysis of C2C12 cell line expressing doxycycline-inducible Ebf1. This experiment was repeated twice (two independent treatments), which showed the same results. **d** Indicated genes were analyzed by RT-qPCR before/after inducing Ebf1 expression. Error bars indicate S.E.M. Two tailed paired Student's $t$ test between untreated and Dox treated cells ($n = 3$ independent treatments). *, $p < 0.05$. **, $p < 0.01$. ***, $p < 0.001$. The exact $p$-values are available in the source file. **e** Single-molecule FISH of indicated marker genes in TA muscle of control or *Ebf1* mutant mice. Arrows indicate myonuclei expressing MTJ-B marker genes. T, tendon. Downregulation of expression was validated in four individuals. Scale bar, 50 μm.

Finally, FISH experiments showed increased expression of MTJ-A marker genes during regeneration, indicating that a higher demand for the products of MTJ genes might exist when the MTJ needs to be re-established, whereas less are needed to maintain this structure.

Another dynamic aspect of myonuclear heterogeneity is demonstrated by our analysis on dystrophic muscles. snRNAseq of *Mdx* muscles revealed a number of compartments not present in uninjured muscle. In particular, we identified the molecular signature of degenerating fibers and a transcriptional program that appears to be associated with fiber repair. These gene signatures might be useful for a quantitative and rapid assessment of muscle damage in the clinic. In addition, many nuclear subtypes present in normal muscle were lost in the *Mdx* muscle, including nuclei expressing the NMJ signature. The protein (but not the transcript) encoded by *Dystrophin* is known to be highly enriched at the NMJ, and the NMJ was previously observed to be functionally abnormal in *Mdx* mice[58–60]. The absence of nuclei that express the NMJ signature in the *Mdx* muscle might provide a molecular correlate for these deficits. Together, our analysis demonstrates that the use of snRNAseq can provide insights into the molecular pathophysiology of muscle disease.

Given the large size and complexity of the muscle tissue, the full diversity of myonuclei likely needs further exploration. Here, we concentrated our analysis on a single muscle group, the tibialis anterior muscle that mainly contains fast fibers, and determined the transcriptional heterogeneity and programs in uninjured, regenerating and dystrophic muscle. An accompanying manuscript[49] successfully used snRNAseq to define nuclear subtypes in the postnatal, adult and aged tibialis anterior muscle. The two studies identified an overlapping set of compartments, but each also found distinct ones, underlining the fact that transcriptional compartments in the muscle are highly dependent on variables like disease or age. Further, different isolation strategies and snRNA sequencing methods were used in the two studies. Our strategy identified the rare MTJ-B or perimysial nuclei subtypes that were not detected by others, and thus the strategy of genetic labeling and isolating myonuclei should be promising for the identification of nuclear subtypes in other muscle groups and contexts. Nevertheless, nuclei from the muscle spindle, a very rare and specialized fiber type, were not detected in any of the datasets that analyzed a random set of myonuclei, but we overcame this limitation by restricting the genetic labeling. The snRNAseq analysis of spindle nuclei revealed many subtypes inside these rare fibers, especially the presence of a specific compartment at the site of innervation by proprioceptive sensory neurons. More generally, our approach should be useful to investigate other syncytial cell types such as the placental trophoblasts or osteoclasts.

## Methods

**Isolation of nuclei from TA muscle.** For each sorting of uninjured, regenerating, or *Mdx* muscles, we pooled two TA muscles from two mice (one TA from each mouse). Dissected TA muscles were minced into small pieces in a 3.5 cm dish on ice with scissors in 300 μl hypotonic buffer (250 mM sucrose, 10 mM KCl, 5 mM MgCl₂, 10 mM Tris-HCl pH 8.0, 25 mM HEPES pH 8.0, 0.2 mM PMSF and 0.1 mM DTT supplemented with protease inhibitor tablet from Roche), and transferred with a 1 ml pipette and cut tip to a 2 ml "Tissue homogenizing CKMix" tube (Bertin instruments KT03961-1-009.2) containing ceramic beads of mixed size. The dishes and tips were washed with 700 μl hypotonic buffer. Samples were incubated on ice for 15 min and homogenized with Precellys 24 tissue homogenizer (Bertin instruments) for 20 sec at 5,000 rpm. Homogenized samples were passed once through 70 μm filter (Sysmex), twice through 20 μm filter (Sysmex), and once though 5 ml filter cap FACS tube (Corning 352235). DAPI (Sigma) was added to final concentration of 300 nM to label DNA. GFP and DAPI double-positive nuclei were sorted using ARIA Sorter III (BD) and BD FACSDiva software 8.0.1. The contour plots were generated using FlowJo (version 10).

For isolation of muscle spindle nuclei, we employed two different protocols. In the first protocol (protocol 1 in Supplementary Fig. 12a), we used 6 TA muscles

from 3 mice (3 TA muscles per 2 ml homogenizer tube) and used the same procedure as described above. From this, we isolated 96 nuclei. For the second procedure (protocol 2 in Supplementary Fig. 12a), we used 8 TA muscle from 4 mice (4 TA muscles per 2 ml homogenizer tube) and aimed to shorten the isolation time. For this, 0.1% Triton X-100 was added to the hypotonic buffer to solubilize the tissue debris, which are otherwise detected as independent particles during FACS. After homogenization and filtration, nuclei were pelleted by centrifuging at 200 g for 10 min at 4 °C. After aspirating the supernatant, the pellet was resuspended in 300 μl hypotonic buffer (without detergent) and passed through the FACS tube. The subsequent FACS sorting yielded 192 spindle nuclei.

**Library generation and sequencing.** 96-well plates for sorting were prepared using an automated pipetting system (Integra Viaflo). Each well contained 1.2 μl of master mix (13.2 μl 10% Triton X-100, 25 mM dNTP 22 μl, ERCC spike-in 5.5 μl and ultrapure water up to 550 μl total) and 25 ng/μl barcode primers. Plates were stored at −80 °C until use.

After sorting single nuclei into the wells, plates were centrifuged at 4000 g for 1 min, incubated on 65 °C for 5 min, and immediately cooled on ice. Subsequent library generation was performed using the CEL-Seq2 protocol as described[26]. After reverse transcription and second-strand synthesis, products of one plate were pooled into one tube and cleaned up using AMPure XP beads (Beckman Coulter). After in vitro transcription and fragmentation, aRNA was cleaned up using RNAClean XP beads (Beckman Coulter) and eluted in 7 μl ultrapure water. 1 μl of aRNA was analyzed on Bioanalyzer RNA pico chip for a quality check. To construct sequencing library, 5 μg aRNA was used for reverse transcription (Superscript II, Thermofisher) and library PCR (Phusion DNA polymerase, Thermofisher). After clean up using AMPure XP beads, 1 μl sample was run on Bioanalyzer using a high sensitivity DNA chip to measure size distribution, which demonstrated the presence of a peak of around 400 bp length. Next-generation sequencing was performed using Illumina HiSeq2500 or NextSeq500 High Output. Further information on sequencing platforms and multiplexing are available in Supplementary Data 5.

**Bioinformatics analysis.** Single-nucleus RNA-Sequencing data was processed using PiGx-scRNAseq pipeline - a derivative of a CellRanger pipeline, but enabling deterministic analysis reproducibility (version 1.1.6)[61]. In short, polyA sequences were removed from the reads. The reads were mapped to the genome using STAR[62]. Number of nuclei, for each sample, was determined using dropbead[63]. Finally, a combined digital expression matrix was constructed, containing all sequenced experiments.

Digital expression matrix post-processing was performed using Seurat (ver 3.1)[64]. The raw data were normalized using the NormalizeData function. The expression of each nucleus was then normalized by multiplying each gene with the following scaling factor: 10,000/(total number of raw counts), log(2) transformed, and subsequently scaled. Number of detected genes per nucleus was regressed out during the scaling procedure.

Variable genes were defined using the FindVariableGenes function with the default parameters. Samples were processed in three groups with differing parameters. Samples originating from uninjured, 7 d.p.i. and 14 d.p.i. were processed as one group, samples from the *Mdx* mouse as the second group and muscle spindle nuclei as the third group. The samples originating from different biological sources contained markedly different properties—number of detected genes and UMIs, which precluded their analysis with the same parameter set.

To test stress response in our dataset, we used the signature of stress-induced genes identified previously[23]. We tested whether the stress-induced genes were co-expressed in individual cells using two different algorithms, AddModuleScore from Seurat and AUCell. The distribution of obtained scores was similar regardless of the algorithms used. Based on this, we concluded that there were only a handful nuclei (less than 10 among all the nuclei analyzed in total) which showed co-expression of known stress genes, and could therefore be considered "stressed".

For samples of the first group, nuclei with less than 500 detected genes were filtered out. Subsequently, genes which were detected at least in 5 nuclei were kept for further analysis. To remove the putative confounding effect between time of sample preparation and biological variable (injury), the processed expression matrices were integrated using the FindIntegrationAnchors function with reciprocal PCA, from the Seurat package. The function uses within batch covariance structure to align multiple datasets,

The integration was based on 2000 top variable features, and first 30 principal components. UMAP was based on the first 15 principal components. Outlier cluster detection was done with dbscan (https://doi.org/10.18637/jss.v091.i01.), with the following parameters eps = 3.4, minPts = 20.

*Mdx* samples contain dying fibers, which have few detected transcripts, as well as others that resemble uninjured fibers. Thus, *Mdx* nuclei show a big variance in the number of detected genes. Therefore, *Mdx* samples were processed by filtering out all nuclei with less than 100 detected genes. Top 100 most variable genes were used for the principal component analysis. UMAP and Louvain clustering were based on the first 15 principal components. Resolution parameter of 1 was used for the Louvain clustering.

In muscle spindle nuclei, we detected fewer genes than in uninjured fibers, but the variance was low. Spindle cell samples were processed by filtering out all nuclei

with less than 300 detected genes. The top 200 most variable genes were used for the principal component analysis. UMAP and Louvain clustering were based on the first 15 principal components. Resolution parameter of 1 was used for the Louvain clustering.

For all datasets, multiple parameter sets were tested during the analysis, and the choice of parameters did not have a strong influence on the results and the derivative biological conclusions. Genes with cluster-specific expression were defined using Wilcox test, as implemented in the FindAllMarkers function from the Seurat package. Genes that were detected in at least 25% of the cells in each cluster were selected for differential gene expression analysis.

NMJ Nuclei Definition: NMJ nuclei were identified based on the expression of three previously known markers (Prkar1a, Chrne, and Ache), using the AddModuleScore function from the Seurat package. All cells with a score greater than 1 were selected as NMJ positive cells. NMJ marker set was expanded by comparing the fold change of gene expression in averaged NMJ positive to NMJ negative cells.

MTJ A and B Nuclei Definition: The original gene sets were extracted from cluster-specific genes detected in the uninjured, 7 d.p.i. and 14 d.p.i. experiment. Cells were scored as MTJ A/B using the aforementioned gene set, with the AddModuleScore function. All cells which had a respective score greater than 1 were labeled as MTJ A/B positive cells.

Mdx nuclei scoring by uninjured and regenerating signatures: First, gene signatures specific for each time point were selected using the FindAllMarkers function from the Seurat library, using the default parameters. Mdx samples were scored using the top 25 genes per time point with the AUCell method (https://doi.org/10.1038/nmeth.4463).

Repetitive element annotation was downloaded from the UCSC Browser database (https://doi.org/10.1093/nar/gky1095). Pseudo-bulk bigWig tracks were constructed for each cluster in uninjured, 7.d.p.i. and 14.d.p.i. The tracks were normalized to the total number of reads. Repetitive element expression was quantified using the ScoreMatrixBin function from the genomation (https://doi.org/10.1093/bioinformatics/btu775) package, which calculates the average per-base expression value per repetitive element. The expression was finally summarized to the repetitive element family (class) level by calculating the average expression of all repeats belonging to the corresponding family (class).

Transcription factor compendium, used in all analyses was downloaded from AnimalTFDB (https://doi.org/10.1093/nar/gky822). The expression map (Fig. 5d) for spindle fibers was created using the DotPlot from the Seurat package on a selected set of cluster-specific genes.

Gene ontology analysis was performed using the Enrichr program (http://amp.pharm.mssm.edu/Enrichr/). The online tool for interactive exploration of the single-cell data—MyoExplorer, was set up using iSEE (https://doi.org/10.12688/f1000research.14966.1).

**Mouse lines and muscle injury.** All experiments were conducted according to regulations established by the Max Delbrück Centre for Molecular Medicine and LAGeSo (Landesamt für Gesundheit und Soziales), Berlin. Mice were housed in constant ambient temperature (23 °C), humidity (56%), and light-night cycle (light on 6:00 am and light off 6:00 pm). HSA-Cre (006149), Calb1-IRES-Cre (028532) and Mdx (001801) mice were obtained from the Jackson laboratory. Rosa26-Lsl-H2B-GFP reporter line was a kind gift from Martyn Goulding (Salk Institute) and was described[65]. For experiments regarding uninjured and regenerating muscles, homozygous Rosa26-LSL-H2B-GFP mice with heterozygous HSA-Cre were used. For Calb1-IRES-Cre and Mdx experiments, the H2B-GFP allele was heterozygous. Mice were in C57/BL6 background and nuclei were isolated from muscle of 2.5-month-old mice. Genotyping was performed as instructed by the Jackson laboratory. For genotyping of the Rosa26-Lsl-H2B-GFP reporter, the following primers were used. Rosa4; 5′-TCA ATGGGCGGGGGTCGTT-3′, Rosa10; 5′-CTCTGCTGCCTCCTGGCTTCT-3′, Rosa11; 5′-CGAGGCGGATCA-CAAGCAATA-3′. Ebf1 mutants[66] and Dysferlin mis-sense mutants[67] were described. To induce muscle injury, 30 μl of cardiotoxin (10 μM, Latoxan, Porte les Vaence, France) was injected into the tibialis anterior (TA) muscle. Further information on mouse conditions are summarized in Supplementary Data 5.

**Preparation of tissue sections.** Freshly isolated TA muscles were embedded in OCT compound and processed as previously described[68]. Frozen tissue blocks were sectioned to 12–16 μm thickness, which were stored at −80 °C until future use.

**Single-molecule FISH (RNAscope).** Otherwise specifically indicated as "covential FISH" in the figure legends, all the FISH experiments were single-molecule FISH using RNAscope. RNAscope_V2 kit was used according to the manufacturer's instructions (ACD/bio-techne). We used Proteinase IV. When combined with antibody staining, after the last washing step of RNA Scope, the slides were blocked with 1% horse serum and 0.25% BSA in PBX followed by primary antibody incubation overnight at 4 °C. The subsequent procedures were the same as regular immunohistochemistry. Slides were mounted with ProLong Gold Antifade mounting solution (Thermofisher). The following probes were used in this study; Smox (559431), Mettl21c (566631), Pdk4 (437161), Ttn (483031), Rian (510531); also synthesized in c2, Vav3 (437431), Col6a3 (552541), Egr1 (423371), Gm10800

(479861), Myh2 (452731-c2) and Calcrl (452281). The following probes were newly designed; Arrdc2 (c1), Cish (c1), Nmrk2 (c1), Slc26a2 (c1), Prkar1a (c1), Tigd4 (c1), Muc13 (c1), Gssos2 (c1), Flnc (c1), Klhl40 (c1), Myh1 (c1), Col22a1 (c2), Ufsp1 (c2), Gm10801 (c2), Xirp1 (c2), Fst1l (c2), human Flnc (c2), Ablim2 (c3), Myh2 (c3) and human Xirp1 (c3).

**Preparation of conventional FISH probes.** Probes of 500–700 bp length spanning exon-exon junction parts were designed using the software in NCBI website. Forward and reverse primers included Xho1 restriction site and T3 promoter sequence, respectively. cDNA samples prepared from E13.5-E14.5 whole embryos were used to amplify the target probes using GO Taq DNA polymerase (Promega). PCR products were cloned into pGEM-T Easy vector (Promega) according to manufacturer's guideline, and the identity of the inserts was confirmed by sequencing. 2 μg of cloned plasmid DNA was linearized, 500 ng DNA was subjected to in vitro transcription with T3 polymerase and DIG- or FITC- labeled ribonucleotides (All Roche) for 2 h at 37 °C. Synthesized RNA probes were purified using RNeasy kit (Qiagen). Probes were eluted in 50 μl ultrapure water (Sigma), and 50 μl formamide was added. We checked the RNA quality and quantity by loading 5 μl RNA to 2% agarose gel. Until future use, probes were stored in −80 °C. The annealing sequences of the FISH probes used in this study are available in Supplementary Data 6.

**Conventional FISH and immunohistochemistry.** The basic procedure for conventional FISH was described before[68] with minor modifications to use fluorescence for final detection. After hybridizing the tissue sections with DIG-labeled probes, washing, RNase digestion, and anti-DIG antibody incubation, amplification reaction was carried out using TSA-Rhodamine (1:75 and 0.001% $H_2O_2$). After washing, slides were mounted with Immu-Mount (Thermo Scientific). When applicable, GFP antibody was added together with anti-DIG antibody (Roche, 1:1000).

When conducting double FISH, the tissue was hybridized with DIG- and FITC-labeled probes, after detection of the DIG signal, slides were treated with 3% $H_2O_2$ for 15 min and then with 4% PFA for one hour at room temperature to eliminate residual peroxidase activity. The second amplification reaction was performed using anti-FITC antibody (Roche, 1:1000) and TSA-biotin (1:50), which was visualized using DyLight 649-conjugated streptavidin (Jackson Immunoresearch, 1:5000).

Antibodies used for this study were: GFP (Aves labs, 1:500), Col3 (Novus, 1:500), ColIV (Millipore, 1:500), CD31-PE (Biolegend, 1:200), F4/80 (Abcam ab6640, 1:500), Laminin (Sigma L9393, 1:500), Egr2 (homemade, 1:2000) and Ebf1 (homemade, 1:50). For Pax7 (homemade, 1:50), we used an antigen retrieval step. For this, after fixation and PBS washing, slides were incubated in antigen retrieval buffer (diluted 1:100 in water; Vector) pre-heated to 80 °C for 15 min. Slides were washed in PBS and continued at permeabilization step. Cy2-, Cy3- and Cy5-conjugated secondary antibodies were all purchased from Jackson Immunoresearch and used at 1:5000.

**FISH experiments using isolated single fibers.** We isolated single extensor digitorum longus (EDL) muscle fibers as described before[69]. Isolated EDL fibers were immediately fixed with 4% PFA and were subjected hybridization in 1.5 ml tubes. After DAPI staining, fibers were transferred on slide glasses and mounted. Spindle fibers were vulnerable to collagenase treatment. Thus, we pre-fixed the EDL tissue and peeled off spindles under the fluorescent dissecting microscope (Leica).

**Acquisition of fluorescence images.** Fluorescence was visualized by laser-scanning microscopy (LSM700, Carl-Zeiss) using Zen 2009 software. Images were processed using ImageJ (ver 1.53a) and Adobe Photoshop 2020, and assembled using Adobe Illustrator 2020.

**Cell culture.** C2C12 cell line was purchased from ATCC, and cultured in high glucose DMEM (Gibco) supplemented with 10% FBS (Sigma) and Penicilllin-Streptomycin (Sigma). To engineer C2C12 cells with doxycycline (Sigma) inducible Ebf1, mouse Ebf1 cDNA (Addgene) was cloned into pLVX Tet-One Puro plasmid (Clontech), packaged in 293 T cells (from ATCC) using psPAX2 and VsvG (Addgene), followed by viral transduction to C2C12 cells with 5 μg/μl polybrene (Millipore). Transduced cells were selected using 3 μg/μl puromycin (Sigma).

**Western blotting.** Cell pellets were resuspended in NP-40 lysis buffer (1% NP-40, 150 mM NaCl, 50 mM Tris-Cl pH 7.5, 1 mM MgCl2 supplemented with protease (Roche) and phosphatase (Sigma) inhibitors), and incubated on ice for 20 min. Lysates were cleared by centrifuging in 16,000 g for 20 min at 4 °C. Protein concentration was measured by Bradford assay (Biorad), and lysates were boiled in Laemmli buffer with beta-mercaptoethanol for 10 min. Denatured lysates were fractionated by SDS-PAGE, transferred into nitrocellulose membrane, blocked with 5% milk and 0.1% Tween-20 in PBS, and incubated overnight in 4 °C with primary antibodies diluted in 5% BSA and 0.1% Tween-20 in PBS. After three times washing with PBST, membranes were incubated with secondary antibodies diluted

in blocking solution for one hour at room temperature. After PBST washing, membranes were developed with prime ECL (Amarsham) and imaged using Chemi-Capt 5000 software. The antibodies used for this study were β-actin (Cell Signaling, 1:1000), ColVI (Abcam, 1:2000) and Ebf1 (homemade, 1:1000). HRP-conjugated secondary antibodies were purchased from Cell Signaling and used at 1:5000.

**RT-qPCR.** Cell pellets were resuspended in 1 ml Trizol (Thermofisher). RNA was isolated according to the manufacturer's guideline. 1 μg of isolated RNA and random hexamer primer (Thermofisher) were used for reverse transcription using ProtoSciprt II RT (NEB). Synthesized cDNA was diluted five times in water, and 1 μl was used per one qPCR reaction. qPCR was performed using 2X Sybr green mix (Thermofisher) and CFX96 machine (Biorad). The data were obtained using Bio-rad CFX manager 3.1 software and calculated with Microsoft Excel 365. The graphs were generated using GraphPad Prism 5. We used β-actin for normalization. Primers were selected from the 'Primer bank' website (https://pga.mgh.harvard.edu/primerbank/). The RT-qPCR primers used in this study are available in Supplementary Data 6.

**Human biopsies.** Human muscle biopsy specimens were obtained from M. vastus lateralis. We selected wheelchair-bound patients with confirmed *DMD* or *DYSF* mutations and severe dystrophic myopathological alterations defined by histology of biopsies. The exact mutation, gender, and age of the participants are summarized at the end of Supplementary Information. The tissues were snap frozen under cryoprotection. Research use of the human material was approved by the ethical committee (EA1/203/08, EA2/051/10, EA2/175/17) at the Charité, Universitätsmedizin Berlin, Germany (https://ethikkommission.charite.de/). Informed consent was obtained from the donors, and the study design and conduct complied with all relevant regulations regarding the use of human study participants.

**Reporting summary.** Further information on research design is available in the Nature Research Reporting Summary linked to this article.

## Data availability

The next-generation sequencing datasets generated in this study are available in the ArrayExpress under accession numbers E-MTAB-8623. The raw DGE count matrix in loom format is available in "http://bimsbstatic.mdc-berlin.de/akalin/MyoExplorer/mm10_UMI.loom". Preliminary DGE matrix as a Seurat object is available in "http://bimsbstatic.mdc-berlin.de/akalin/MyoExplorer/mm10.Seurat.RDS". Also, we provide an interactive webpage where the users can explore their gene of interest (https://shiny.mdc-berlin.de/MyoExplorer/). Uncropped Western blot images for Fig. 6c are provided in Supplementary Fig. 14. The ENCODE project dataset used for Fig. 6b is available under the accession code ENCSR000DZQ. The database for repeat elements in Supplementary Fig. 7 was downloaded from UCSC Browser (https://doi.org/10.1093/nar/gky1095). Transcription factor compendium for Fig. 6a was downloaded from Animal TFDB (https://doi.org/10.1093/nar/gky822). Other data that support the findings of this study are available from the corresponding authors upon reasonable requests. Source data are provided with this paper.

## Code availability

The codes used in this study are accessible in GitHub server (https://github.com/BIMSBbioinfo/MyoExplorer).

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

## Acknowledgements

The authors thank Dr. K. Song for advice and protocols on snRNAseq and Dr. T. Muller for critical reading of the manuscript, C. Paeseler and P. Stallerow for help with animal care (all MDC), as well as the MDC core facilities for flow cytometry (led by Dr. Hans-Peter Rahn) and next-generation sequencing (led by Dr. Sascha Sauer). We are grateful to Dr. Martyn Goulding (Salk Institute) for providing the H2B-GFP reporter mouse line, and Drs. M. Derecka and R. Grosschedl (MPI for Immunobiology and Epigenetics) for providing Ebf1 tissues and reagents. This work was supported by the Helmholtz Association (C.B) and AVH postdoctoral fellowship (M.K).

## Author contributions

M.K. and C.B. conceived the work and designed the project. M.K. led the project, performed the experiments and analyzed the data. V.F. and A.A. performed the bioinformatics analysis. B.B. contributed to the experiments, especially generated the sequencing library. E.D.L. also helped with library generation. V.S and S.S. collected and prepared human patient biopsies and *dysferlin* mutant mouse samples. M.K. and C.B. wrote the manuscript with comments by all authors.

## Funding

## Competing interests

The authors declare no competing interests.
