## [Peer Review File · Nature Communications]

Reviewers' Comments:

Reviewer #1:

Remarks to the Author:

In the revision of the manuscript, Minchul Kim et al. addressed some of the points suggested by the reviewers. Importantly, they sequenced another 1,200 nuclei from uninjured muscle. This permitted a better clusterization of the myonuclei. Although, their data are supported by FISH experiments, it is not clear for all the proposed types of myonuclei their functions, raising the question about their true biological significance (e.g. cluster 8 of the original manuscript is here proposed that it was only a computational artifact). Some clusters are labelled with the name of their marker genes, and their functions in the muscle fibers are only superficially suggested. To reach the standard level of the journal, the biological roles of these clusters of myonuclei should be verified and explored in greater depth, otherwise the paper remains descriptive with limited mechanistic insight.

Reviewer #2:

Remarks to the Author:

The authors have addressed all my comments adequately.
I congratulate them for a very solid and interesting study.

Reviewer #3:

Remarks to the Author:

The authors have addressed my concerns. Thank you.

Reviewer #4:

Remarks to the Author:

I commend the effort the authors have made to address those comments and I'm satisfied with their response.

We thank all the reviewers for taking their time and efforts to review our manuscript. We are pleased to see that the reviewers were overall satisfied with our revised manuscript.

Reviewer #1 (Remarks to the Author):

In the revision of the manuscript, Minchul Kim et al. addressed some of the points suggested by the reviewers. Importantly, they sequenced another 1,200 nuclei from uninjured muscle. This permitted a better clusterization of the myonuclei. Although, their data are supported by FISH experiments, it is not clear for all the proposed types of myonuclei their functions, raising the question about their true biological significance (e.g. cluster 8 of the original manuscript is here proposed that it was only a computational artifact). Some clusters are labelled with the name of their marker genes, and their functions in the muscle fibers are only superficially suggested. To reach the standard level of the journal, the biological roles of these clusters of myonuclei should be verified and explored in greater depth, otherwise the paper remains descriptive with limited mechanistic insight.

We agree with the reviewer that the paper is descriptive. Nevertheless, the existence of the new populations is supported by our FISH experiments; their biological significance is supported by very specific anatomical locations (e.g. perimysium, dying fibers or spindle clusters), other people's recent work (e.g. by groups of Delphine Duprez and Peleg Hasson on MTJ-B; their manuscripts are posted in bioRxiv), and previous literatures showing the importance of the top marker genes in muscle physiology (e.g. Rian+ or fiber repair nuclei). There are few clusters that are more enigmatic. In general, an in-depth functional analysis of these compartments requires further work. Our paper paves the strategy on how this could be achieved.

Taking this reviewer's comments into consideration, we added a sentence into the discussion that reads 'The functional role of the nuclear populations will need further characterization in the future.' (lines 335-336).

Reviewer #2 (Remarks to the Author):

The authors have addressed all my comments adequately. I congratulate them for a very solid and interesting study.

We thank this reviewer for the positive evaluation and for her/his comments on our original manuscript, which greatly strengthened our paper.

Reviewer #3 (Remarks to the Author):

The authors have addressed my concerns. Thank you.

We thank this reviewer for the positive evaluation and for her/his comments on our original manuscript, which greatly strengthened our paper.

Reviewer #4 (Remarks to the Author):

I commend the effort the authors have made to address those comments and I'm satisfied with their response.

We thank this reviewer for the positive evaluation and for her/his comments on our original manuscript, which greatly strengthened our paper.